# An Inhaled Nanoemulsion Encapsulating a Herbal Drug for Non-Small Cell Lung Cancer (NSCLC) Treatment

**DOI:** 10.3390/pharmaceutics17050540

**Published:** 2025-04-22

**Authors:** Mural Quadros, Mimansa Goyal, Gautam Chauhan, Dnyandev Gadhave, Vivek Gupta

**Affiliations:** Department of Pharmaceutical Sciences, College of Pharmacy and Health Sciences, St. John’s University, 8000 Utopia Parkway, Queens, NY 11439, USA

**Keywords:** nanoemulsion, non-small cell lung cancer, Celastrol, inhalation, next generation impactor

## Abstract

**Background:** Celastrol (Cela), a phytochemical extracted from *Tripterygium wilfordii*, has been extensively investigated for its potential anti-inflammatory, anti-psoriatic, antioxidant, neuroprotective, and antineoplastic properties. However, its clinical translation is limited due to poor bioavailability, low solubility, and nonspecific toxicity. This study aimed to develop and evaluate an inhalable Cela-loaded nanoemulsion (NE) formulation to enhance targeted drug delivery and therapeutic efficacy in non-small cell lung cancer (NSCLC). **Methods:** The NE formulation was optimized using Capmul MCM (25%), Tween 80 (20%), Transcutol HP (5%), and water (50%) as the oil, surfactant, co-surfactant, and aqueous phase, respectively. Physicochemical characterization included globule size, zeta potential, and drug release in simulated lung fluid. In vitro aerosolization performance, cytotoxicity in NSCLC cell lines (A549), scratch and clonogenic assays, and 3D tumor spheroid models were employed to assess therapeutic potential. **Results:** The NE showed a globule size of 201.4 ± 3.7 nm and a zeta potential of −15.7 ± 0.2 mV. Drug release was sustained, with 20.4 ± 5.5%, 29.1 ± 10%, 64.6 ± 4.1%, and 88.1 ± 5.2% released at 24, 48, 72, and 120 h, respectively. In vitro aerosolization studies indicated a median aerodynamic particle size of 4.8 ± 0.2 μm, confirming its respirability in the lung. Cell culture studies indicated higher toxicity of NE-Cela in NSCLC cells. NE-Cela significantly reduced A549 cell viability, showing a ~6-fold decrease in IC_50_ (0.2 ± 0.1 μM) compared to Cela alone (1.2 ± 0.2 μM). Migration and clonogenic assays demonstrated reduced cell proliferation, and 3D spheroid models supported its therapeutic activity in tumor-like environments. **Conclusions:** The inhalable NE-Cela formulation improved Cela’s physicochemical limitations and demonstrated enhanced anti-cancer efficacy in NSCLC models. These findings support its potential as a targeted, well-tolerated therapeutic option for lung cancer treatment.

## 1. Introduction

Over the past few decades, several phytoconstituents discovered in Chinese medicinal herbs have been investigated for treating various diseases, ranging from inflammation to cancer [1,2,3,4,5]. One such example is Celastrol (Cela), a well-known photochemical extracted from *Tripterygium wilfordii*, which has been extensively studied for its anti-inflammatory, anti-psoriatic, antioxidant, and neuroprotective properties [6,7,8]. Cela has also been explored for its anticancer activity in several cancers, such as osteosarcoma, melanoma, colon, and lung cancer [9]. Cela is known to increase the reactive oxygen species (ROS) levels in cancerous cells, initiating endoplasmic reticulum stress and influencing cell apoptosis, while the elevated intracellular ROS levels can further exert an antineoplastic effect by inhibiting the activation of signaling pathways, eventually leading to cell apoptosis [10,11,12].

Among all cancer types, lung cancer is the leading cause of cancer-related deaths worldwide, accounting for 1.8 million deaths annually [13]. Lung cancer is further classified based on the histology of cancer cells as non-small cell lung cancer (NSCLC; 85%) and small cell lung cancer (SCLC; 10–15%). **NSCLC is the more prominent form of lung cancer,** causing proliferation in various epithelial cells lining the lungs and with a 5-year survival rate of only 23% [14]. Conventional treatment for NSCLC includes surgical resection of the lung or a region of the lung followed by intense chemotherapy and radiotherapy. In recent years, immunotherapy and targeted therapies have emerged as promising approaches for NSCLC, offering greater treatment precision. Immunotherapy leverages the body’s immune system to recognize and attack cancer cells, with immune checkpoint inhibitors such as pembrolizumab and nivolumab successfully targeting proteins like PD-1 and PD-L1 to enhance the immune response against cancer cells [15,16]. Targeted therapies, particularly tyrosine kinase inhibitors (TKIs) such as Osimertinib and erlotinib, have also become prominent in treating NSCLC, especially in cases with specific genetic mutations like EFGR [17,18]. These therapies allow for a more focused approach by blocking molecular pathways essential for cancer cell survival and growth, thus reducing the widespread impact on healthy cells compared to traditional chemotherapy [19].

Despite significant advancements in cancer treatment modalities, including chemotherapy, radiotherapy, immunotherapy, and targeted therapy, each approach is associated with its limitations that serve as a bottleneck to achieving high efficacy [20]. Chemotherapy is limited by high-dose regimens and nonspecific targeting, leading to severe adverse effects [21]. Radiation therapy, while effective, often affects surrounding healthy tissues, resulting in off-target toxicity [22]. Immunotherapy, although promising, is limited by variable patient response and immune-related adverse effects [23]. Similarly, targeted therapies are challenged by the development of acquired resistance and reduced efficacy in mutated tumors [24]. These limitations collectively contribute to treatment-related toxicity and play a substantial role in the high mortality rates associated with cancer. However, despite these challenges, small-molecule chemotherapeutic agents continue to play a pivotal role as a cornerstone of cancer therapy [25].

Celastrol (Cela), a plant-derived compound, has gained attention as a potential treatment for NSCLC due to its ability to induce apoptosis and inhibit proliferation by elevating reactive oxygen species (ROS), inducing endoplasmic reticulum stress, disrupting cellular homeostasis, and suppressing the STAT3 signaling pathway [12]. Although Cela offers a wide range of therapeutic targets, it also exhibits physicochemical and physiological limitations such as a narrow therapeutic window, low bioavailability (17.06%), poor water solubility (13.3 ± 0.8 µg/mL), and off-target effects in the liver and heart [26,27,28]. Moreover, according to the biopharmaceutical classification system (BCS), Cela is classified as a BCS class IV molecule with low solubility and permeability properties. These physicochemical and pharmacokinetic limitations of Cela pose a bottleneck for its clinical translation. Several strategies have been implemented to circumvent the physicochemical and pharmacokinetic limitations of Cela, such as structural derivatization [29], formation of water-soluble cyclodextrin complexes [30], and encapsulation in nano/micro drug delivery systems [30,31]. Additionally, recent studies have explored the use of smart nanocarriers to encapsulate herbal drugs; for instance, in a recent study, Lai et al. encapsulated paclitaxel in iron nanoparticles for magnetic resonance imaging-guided anti-tumor therapy [32]. These approaches can be explored to enhance Cela’s stability, improve its solubility, and prolong its circulation time in the body, thereby optimizing its efficacy for potential therapeutic applications in NSCLC.

Lipid-based nanoemulsions (NEs) have emerged as a promising drug delivery system, specifically designed to enhance the solubility of poorly soluble therapeutic agents and improve their stability and efficacy. Among various nanocarriers, including liposomes, solid lipid nanoparticles, and polymeric nanoparticles, NEs are particularly effective for delivering hydrophobic drugs [33]. In simple terms, NEs comprise two immiscible phases—an oil phase and an aqueous phase—stabilized by amphiphilic surfactant molecules that facilitate their formation. Depending on the arrangement of the dispersed and continuous phases, NEs can be classified into oil-in-water (O/W) or water-in-oil (W/O) types, allowing for tailored applications in drug formulation and delivery [34]. Hydrophobic drugs are typically formulated as O/W emulsions, generally with a mean droplet diameter of less than 500 nm. Given these characteristics, NEs present an effective means of delivering Cela, addressing its challenges related to poor solubility and enhancing its therapeutic potential. Furthermore, administering the NE formulation via the pulmonary route enables targeted delivery directly to the tumor site within the lungs [35]. This approach enhances local drug deposition at the NSCLC tumor site, optimizing therapeutic efficacy while minimizing systemic exposure and associated side effects, with the added benefit of enhanced patient compliance, making it a promising approach for treating NSCLC [36,37].

In this study, we highlight the potential of inhaled nanoemulsion-encapsulated celastrol (NE-Cela) as an innovative therapeutic strategy for enhancing the treatment of non-small cell lung cancer (NSCLC). We hypothesize that encapsulating Cela in a nanoemulsion will significantly improve its therapeutic efficacy by overcoming its poor solubility and facilitating targeted delivery to lung tissues via the inhalation route. By optimizing the composition of NE-Cela, we aim to achieve a stable formulation that maximizes local drug concentrations, thereby enhancing the overall anticancer activity while minimizing systemic side effects.

## 2. Materials and Methods

### 2.1. Materials and Chemicals

Celastrol (>98% purity) was purchased from Adooq Bioscience (Irvine, CA, USA). Capmul MCM was procured as a gift sample from Abitech Corp (Janesville, WI, USA). Transcutol HP was procured as a gift sample from Gattefosse (Paramus, NJ, USA). Tweens 80 was purchased from Spectrum (New Brunswick, NJ, USA). 3-(4,5-Dimethylthiazol-2-yl)-2,5-diphenyltetrazolium bromide (MTT), LC-MS grade water, acetonitrile (ACN), dimethyl sulfoxide (DMSO), and methanol were purchased from Fisher Scientific (Hampton, NH, USA). Milli-Q water was used as a formulation vehicle. All other chemicals and kits were purchased from third-party vendors and are listed in respective sections.

### 2.2. Cell Culture

NSCLC cell lines A549 and H1299, as well as the human embryonic kidney (HEK-293) cell line and normal primary human airway epithelial (AEC) cells, were procured from ATCC (Manassas, VA, USA). H1299 and A549 cells were cultured in RPMI-1640 medium (Corning, NY, USA), while HEK-293 cells were cultured in DMEM media (Corning, NY, USA). Both culture media were enriched with 10% heat-inactivated fetal bovine serum (FBS; Atlanta Biologicals, Minneapolis, MN, USA), 5% sodium pyruvate (Corning Inc., Corning, NY, USA), and 5% penicillin-streptomycin (Corning Inc., Corning, NY, USA). AEC cells were cultivated in an airway epithelial basal medium supplemented with a bronchial epithelial cell growth kit purchased from ATCC (Manassas, VA, USA). All cell lines were incubated at 37 °C with 5% CO_2_. Cells were allowed to grow to 85–90% confluency before being either subcultured or used in experiments.

### 2.3. Quantification of Cela Through Ultra-Performance Liquid Chromatography (UPLC)

A Waters Acquity reverse-phase UPLC (Waters, Milford, MA, USA) method for the quantification of Cela was previously developed in our lab [30]. The detailed methodology is provided in the Appendix A.

### 2.4. Preparation of Nanoemulsion (NE) and Construction of Pseudo-Ternary Phase Diagrams

The components of the NE, namely oil, surfactant, and co-surfactants such as Capmul MCM, Tween 80, and Transcutol HP, were selected based on literature for molecules similar to Cela [33,38].The NE was prepared using a spontaneous low-energy method. Briefly, varying amounts of the oil phase, specifically ratios of 1:9, 2:8, 3:7, 4:6, 5:5, 6:4, 7:3, 8:2, and 9:1, were mixed with different surfactant/cosurfactant (S_mix_) ratios of 1:1, 2:1, 3:1, and 4:1. For example, to prepare a 1:9 oil-to-S_mix_ ratio with a 1:1 S_mix_ ratio, 0.1 mL of Capmul MCM was mixed with 0.9 mL of S_mix_ containing a 1:1 ratio of Tween 80 and Transcutol HP. The resulting mixtures were titrated with water dropwise while vortex mixing for 30 s and were visually observed for turbidity and gelation. A pseudo-three-dimensional phase diagram was created with three axes representing water, oil, and S_mix_ ratios. The phase diagrams were constructed using CHEMIX School software (Version 12.2). The ideal compositions from the phase diagrams were used to prepare NE-Cela. Briefly, 10 mg of the drug was mixed with the oil phase and allowed to solubilize overnight, followed by mixing with S_mix_ and water to prepare 2 mL of the formulation. The formulation was further characterized and studied for its anti-cancer activity.

### 2.5. Physicochemical Characterization

#### 2.5.1. Hydrodynamic Size and Surface Charge Analysis

The globule size, polydispersity index (PDI), and zeta potential were performed by the dynamic light scattering (DLS) technique employing a Zeta Sizer (Malvern Instruments, Malvern, UK). The NEs were diluted 20-fold with Milli-Q water for size and charge analysis. The diluted NEs were subjected to particle size analysis at 25 °C with the scattering angle of measurement at 173°. The samples were placed in a DTS 1070 zeta cell for surface charge analysis.

#### 2.5.2. Transmission Electron Microscopy

The globule size and morphology were determined using transmission electron microscopy (TEM) [39]. The detailed methodology is provided in the Appendix A.

#### 2.5.3. Drug Entrapment Efficiency

The amount of drug entrapped in the NE globules was quantified using the UPLC method described in Section 2.3. Briefly, Cela-NE was subjected to centrifugation at 10,000 rpm for 15 min to separate the unentrapped drug. The supernatant was diluted with acetonitrile (ACN) to break the emulsion and dissolve the entrapped Cela in ACN. The resulting sample was centrifuged at 10,000 rpm for 15 min to separate precipitated emulsion components, and the clear solution was subsequently injected into the UPLC system for analysis. The entrapment efficiency (%EE) of the NE-Cela was calculated using the following equation.(1)$$\%\;EE=\frac{Entrapped\;Cela}{Weighed\;Cela}\times 100$$

#### 2.5.4. In Vitro Drug Release

In vitro release studies were performed using the tube method to determine the drug release kinetics from Cela-NE. Drug release studies were performed in simulated lung fluid prepared using a published formula [40]. Briefly, 10 μL of the formulation was dispersed in each 990 μL of simulated lung fluid within separate microtubes. A set of microtubes were placed in a temperature-controlled shaker at 37 °C and 180 rpm [41]. Microtubes were withdrawn at predetermined time points of 1, 2, 4, 6, 8, 12, 24, 48, 72, and 128 h. At each time point, one microtube was removed, and the contents were centrifuged at 10,000 rpm for 20 min. The supernatant was examined for Cela content employing the validated UPLC technique depicted in Section 2.3 and Appendix A.

#### 2.5.5. In Vitro Aerosolization

In vitro deposition of NE-Cela in the lungs was studied using Next Generation Impactor^TM^ (NGI; Model 170, NGI: MSP Corp., Shoreview, MN, USA) as used in previous studies by our group [39,42]. The detailed methodology is provided in the Appendix A.

### 2.6. Stability Studies

The stability of the optimized NE was studied at 40 °C and 75% relative humidity (accelerated conditions) in the stability chamber for four weeks. Similarly, some samples were stored at 4 °C for 4 weeks. After four weeks, the samples were analyzed for changes in globule size, zeta potential, and drug content, as discussed in Section 2.5.

### 2.7. Cell Culture Studies

#### 2.7.1. Cytotoxicity Studies

The cytotoxic potential of NE-Cela was evaluated over a concentration range of 0.16 to 5 µM for 48 h using MTT assay in two NSCLC cell lines, i.e., A549 and H1299 cells, following our published studies [43,44]. The IC_50_ was determined by transforming the data to log value and subsequent non-linear curve fitting using Graph Pad Prism Software Version 9.0 (San Diego, CA, USA). The detailed methodology is provided in the Appendix A.

The biocompatibility of plain NE was also determined on HEK-293, a human embryonic kidney cell line, and human alveolar epithelial cells (AEC) using the MTT assay, as described earlier. Both cells were treated with blank-NE concentrations equivalent to 0.04 to 0.6 µM. Following incubation for 48 h, the cells were incubated with MTT and DMSO to determine the percentage of cell viability.

#### 2.7.2. Wound Healing Assay

A wound healing assay, also known as a scratch assay, was performed to understand the treatments’ effect on the cells’ metastatic properties [45]. The detailed methodology is provided in the Appendix A.

#### 2.7.3. Clonogenic Assay

Conventional treatments for cancer, like surgery and chemotherapy, often cannot eliminate all cancer cells. The few remaining cells can grow over a period to cause the reoccurrence of cancer. The clonogenic assay is an in vitro assay that assesses the ability of treatment to inhibit the formation of colonies from single cells. This assay was performed following a previously published protocol [46]. The detailed methodology is provided in the Appendix A.

#### 2.7.4. Spheroid Assay

A three-dimensional tumor microenvironment was created to study the effect and activity of the formulations in solid tumors. The efficacy of Cela-NE treatment was compared against plain Cela in both single and multiple-dose treatment models at IC_50_ and ½ IC_50_ concentrations. Phenotypic efficacy was measured by assessing spheroid volume over a 15-day treatment period. The detailed methodology is provided in the Appendix A.

#### 2.7.5. Live–Dead Assay

A live–dead assay was performed to visualize the live and dead cells in the spheroid mass. The assay was performed on the 15th day using a live–dead assay kit. The detailed methodology is provided in the Appendix A.

### 2.8. Data Representation and Statistical Analysis

The statistical significance was evaluated through Student’s t-test and one-way ANOVA followed by a post hoc Tukey test using GraphPad Prism 9.5.1 (GraphPad, Inc., San Diego, CA, USA). All experiments were performed in triplicates, representing their results as mean ± SD unless stated otherwise. The values reported with *p* < 0.05 and below were considered statistically significant.

## 3. Results

### 3.1. Quantification of Cela by UPLC

The UPLC analytical method for quantification of Cela was successfully established and validated using the method described in Section 2.3. The Waters Acquity series UPLC (Waters, Milford, MA, USA) system was used to quantify Cela. The stationary phase column was Xbridge BEH shield RP18 2.5 μm with a dimension of 3.0 × 100.0 mm. The binary mobile phase (MP) constituted 0.1% orthophosphoric acid and acetonitrile in a ratio of 15:85 *v*/*v*. The flow rate of MP was optimized to 0.5 mL/min. The eluents from the column were detected at wavelength λmax of 425 nm. A sharp quantitative peak was observed at a retention time of 1.2 min. The representative chromatogram overlay (Appendix A) and standard curve (Appendix A) are presented in the Appendix A.

### 3.2. Selection of NE Components and Construction of Pseudo-Ternary Phase Diagrams

Cela demonstrated excellent solubility in Capmul MCM, which was selected as the oil phase for the formulation. Tween 80 and Transcutol HP, known for their compatibility and solubilizing properties, were chosen as the surfactant and co-surfactant, respectively [47,48]. Tween 80, a non-ionic surfactant with a high hydrophilic–lipophilic balance (HLB) value, is widely used in pharmaceutical formulations for its emulsifying properties and safety profile, making it suitable for clinical applications [49]. Following the selection of these components, the next step was to determine their optimal proportions to develop a stable NE. This was achieved using pseudo-ternary phase diagrams, a valuable tool for optimizing the ratios of oil, water, and a fixed combination of surfactant and co-surfactant (S_mix_). In this study, Tween 80 and Transcutol HP were combined in various S_mix_ ratios (1:1, 2:1, 3:1, and 4:1) to construct the pseudo-ternary phase diagrams, aiding in the identification of regions suitable for stable NE formation. Among these, the pseudo-ternary phase diagram generated with a S_mix_ ratio of 4:1 exhibited the largest region of stability (Figure 1) and was, therefore, selected for the formulation of NE. The optimized composition of the NE (Figure 2A) was Capmul MCM (25%), Tween 80 (20%), Transcutol HP (5%), and water (50%) as the oil, surfactant, co-surfactant, and aqueous phase, respectively.

### 3.3. Physicochemical Characterization

The optimized NE-Cela formulation was characterized by its following physiochemical parameters.

#### 3.3.1. Size and Surface Charge

The hydrodynamic size was measured with the principle of dynamic light scattering (DLS), which monitors the variation in light scattering because of the Brownian motion of the globules as a function of time. The globule size of NE-Cela was 201.4 ± 3.7 nm with a PDI of 0.4 ± 0.1, which suggests a moderately monodispersed system. The size and PDI of the NE-Cela were retained at 50 and 100-fold dilution, demonstrating the ability of the nanoemulsion to withstand dilution in the biological fluids (Figure 2B).

Zeta potential is measured from the electrophoretic mobility of the oil droplets, which aids in predicting the dispersion stability of the system. The surface charge analysis of the diluted NE indicated the zeta potential to be −15.7 ± 0.2 mV, suggesting good electrokinetic stability of nanoemulsion from phase separation and coalescence [50] (Figure 2B).

#### 3.3.2. Morphological Assessment with Transmission Electron Microscopy (TEM)

The morphological study of NEs was performed using TEM, where a beam of electrons is incident onto a thin foil of an immobilized specimen. The TEM images (Figure 2C) demonstrated the monodispersed and spherical globules of size 200 nm at 50-fold dilution, which corroborates with size analysis obtained from DLS. Figure 2D depicts images of optimized clear NE, both (a) blank and (b) Cela-loaded NE.

#### 3.3.3. Drug Encapsulation

The amount of drug encapsulated in NE-Cela was calculated using Equation 1 and was estimated to be 97.2 ± 1.8% (Figure 2B). Our studies demonstrated the Cela concentration in the NE formulation to be 4.8 ± 1.2 mg/mL, a significant improvement over the reported Cela’s aqueous solubility of 13.3 ± 0.8 µg/mL (>360-fold solubility enhancement) [28]. The high encapsulation of Cela in the NE can be attributed to its high log P of 5.6, which may increase its affinity to the oil phase [51]. Moreover, incorporating surfactants and co-surfactants, i.e., Tween 80 and Transcutol HP, may further increase the solubility of Cela.

#### 3.3.4. In Vitro Drug Release

The release of Cela from the NE was evaluated using the tube method in simulated lung fluid. The release profile is presented in Figure 2E, which illustrates a sustained release over 5 days, with only 13% of the drug released in the first 2 h. The drug was released from the NE in a time-dependent manner, with 20.4 ± 5.5%, 29.1 ± 10%, 64.6 ± 4.1%, and 88.1 ± 5.2% drug released at 24, 48, 72, and 120 h (5 days), respectively. The release profile was fitted into various release models to understand the kinetic profile of NE-Cela. As evident from the R^2^ value, the zero-order model was found to be the most appropriate model with the highest correlation coefficient value of 0.9462 (Figure 2F), suggesting that the drug is released from the carrier at a constant rate [52]. All release kinetic profile plots are presented in Appendix A.

#### 3.3.5. In Vitro Aerosolization

The deposition of the inhaled NE along the respiratory tract depends on various physiochemical parameters of the NE, such as globule size, surface charge, and lipophilicity. The deposition and aerosolization potential of the NE can be predicted by determining the aerodynamic properties of the vesicles [39]. The MMAD is a crucial parameter that describes the median aerodynamic particle size distribution of an aerosol by mass and was found to be 4.8 ± 0.2 μm (Figure 3C). The geometric standard deviation (GSD), which describes the spread of the aerodynamic particle distribution, was found to be 2.4 ± 0.1. These parameters suggest that most of the emitted dose will be delivered to the respirable region of the lungs. The fine particle fraction (FPF, %), also known as the respirable fraction, was calculated to be 70.7 ± 5.2%, which suggests the excellent aerosolization capability of NE-Cela (Figure 3C). Figure 3A,B represent deposition of NE-Cela at each NGI stage and % cumulative deposition respective to each stage concerning cutoff diameter, respectively. Similarly, Figure 3C exhibits the aerosolization performance of developed nanoformulations, which comply with the standards of aerosolization parameters. Therefore, this study demonstrates excellent inhalable characteristics of NE-Cela for local lung deposition.

### 3.4. Cell Culture Studies

#### 3.4.1. Cytotoxicity Studies

The in vitro cytotoxicity assessment of NE-Cela and Cela was performed using an MTT assay on two NSLC cell lines, A549 and H1299. Figure 4 demonstrates the cytotoxic potential of NE-Cela and Cela. The results showed a significant reduction in IC_50_ values for NE-Cela compared to Cela. For A549 cells, a ~6-fold reduction in IC_50_ values was observed with NE-Cela (0.2 ± 0.1 μM) compared to 1.2 ± 0.2 μM for Cela (*p* < 0.0001) (Figure 4A). Similar results were observed in another human NSCLC immortalized cell line, i.e., H1299, where the IC_50_ value of NE-Cela was decreased to 0.2 ± 0.2 μM from 0.5 ± 0.3 μM of Cela, a significant 2.5-fold reduction (Figure 4B) (*p* < 0.0001). The difference in Cela activity between H1299 and A549 cells may be due to variations in their genetic makeup and sensitivity to treatment. H1299 cells, which lack functional P53, are generally more responsive to Cela than A549 cells [53]. These results indicate that encapsulating Cela into NE enhanced its cytotoxic potential when compared with its plain counterpart (Figure 4C). Furthermore, to make sure that the cytotoxic potential of NE-Cela arose from Cela only and not from the constituents of NE, the biocompatibility of plain NE (non-drug-loaded) was evaluated in HEK-293 (human embryonic kidney cells) and alveolar epithelial cells (AEC) over the range of 0.6 to 0.04 μM equivalent Cela concentration. As shown in Figure 4D, blank NE exhibited 100% cell viability at the concentration beyond or around IC_50_ values of NE-Cela in HEK-293 cells. Although the blank NE displayed slight toxicity in AEC cells, it maintained approximately 80% cell viability, further supporting its biocompatibility in normal cells. These results confirm that NE is biocompatible with normal cells (Figure 4D).

#### 3.4.2. Wound Healing Assay

NSCLC is notoriously known to metastasize to distal organs such as bone, brain, and liver [54]. Once metastasized, cancer worsens the existing condition and causes a steep increase in the mortality rate. Therefore, it is crucial to develop a treatment that has a cytotoxic effect on cancer cells and inhibits cell migration, i.e., metastases. The optimized NE-Cela was evaluated for its impact on the migration of A549 cells. As seen in Figure 5, a scratch was performed at 0 h and was subsequently observed over a period of 48 h after respective treatments. The cellular migration in treatment groups, i.e., NE-Cela and Cela, was compared with that of the control group. Scratch closure (%) was calculated for each treatment group relative to its corresponding 0 h wound area. As can be seen, the inhibition was dose-dependent, with prominent inhibition observed at 24 h for both 0.6 and 1.2 μM concentrations (Figure 5A). Treatment with Cela 0.6 μM exhibited some scratch closure after 24 h (22.6 ± 6.0%), while scratches treated with NE-Cela did not simultaneously show a significant scratch closure, i.e., closure of just 7.2 ± 1.0% (*p* < 0.001) (Figure 5B). Increasing the concentration to 1.2 μM inhibited the scratch closure in both treatments until 24 h, with NE-Cela showing higher (6.6 ± 3.0% closure) inhibition than Cela (16.6 ± 3.9%). At 48 h, Cela demonstrated about 100 ± 1.2% closure compared to that of NE-Cela (24.9 ± 5.2% closure) (*p* < 0.001) (Figure 5A,B). These results demonstrate the significantly improved inhibitory effect of NE-Cela on the migratory properties of A549 cells.

#### 3.4.3. Clonogenic Assay

The clonogenic assay is an essential cell-based tool to understand the inhibitory effect of treatment on the colony-forming ability of single cancer cells. Cela and NE-Cela were studied for their inhibitory effect in single A549 cells. Figure 6A demonstrates representative camera images, highlighting the superior inhibitory effect of NE-Cela over plain Cela at both 0.6 and 1.2 μM concentrations. After a 48 h drug treatment and subsequent 7 day incubation, the % growth of colonies that survived 0.6 μM was 52.2 ± 37.9% for plain Cela in comparison to that of NE-Cela, which was 17.3 ± 2.0%, suggesting the strong colony inhibition ability of NE-Cela (Figure 6B). At the concentration of 1.2 μM, further increase in inhibition was observed for NE-Cela with 13.6 ± 2.8% (*p* < 0.005) growth as compared to its plain counterpart, indicating 39.3 ± 17.1% growth in colonies. These results indicate NE-Cela’s superior ability to inhibit the formation of cancer colonies and, thereby, cancer recurrence.

#### 3.4.4. 3D Spheroid Assay

The in vitro studies were performed to determine the formulation’s effectiveness against a monolayer of cells [55]. These monolayer assays are not analogous to those of physiological tumors as they lack the tumorigenic micro-environment, cellular interaction, spatial architecture, and presence of tumor heterogeneity [56]. In this regard, 3D spheroid models have gained importance as they are a reliable tool for evaluating anti-tumor efficacy [57,58]. This manuscript presents a comparative study to determine the anti-tumor effectiveness of NE-Cela and Cela in 3D spheroid models [44,59].

##### Spheroid Volume

The spheroid images for single-dose and multiple-dose treatment groups were captured over a period of 15 days. Representative spheroid images are presented in Figure 7A (single dose), and Figure 8A (multiple dose). In the single-dose study, i.e., when a single treatment was given on day 3, the spheroids of the control group were found to have higher tumor volumes of 2.9 ± 0.9 mm^3^ on day 15, compared to 1.4 ± 0.36 mm^3^ on day 0, i.e., a two-fold increase in the tumor volume (Figure 7A). The response of the treatment groups was as follows. On day 15, a significant reduction in spheroid volumes was observed for NE-Cela-treated spheroids; i.e., tumor volumes of 1.5 ± 0.1 mm^3^ and 1.4 ± 0.1 mm^3^ at 0.6 and 1.2 μM Cela concentrations, respectively (Table 1). On the other hand, plain Cela-treated spheroids did not respond to either of the Cela concentrations and were similar in volume to that of the control spheroids; i.e., 2.2 ± 0.4 mm^3^ and 1.8 ± 0.3 mm^3^ at 0.6 and 1.2 μM Cela concentrations, respectively (control vs. NE-Cela 1.2 μM *p* < 0.0001; control vs. NE-Cela 0.6 μM *p* < 0.0001; Cela 0.6 μM vs. NE Cela 0.6 μM *p* < 0.005) (Figure 7B). Therefore, in single-dose treatment, the spheroid volumes for NE-Cela at both concentrations were significantly decreased when compared with the plain drug counterpart and the control group.

A similar volume analysis was performed on spheroids with a multidose treatment regimen (Figure 8B). The spheroids in the control group were measured to have volumes of 1.6 ± 0.3 mm^3^ and 2.7 ± 0.4 mm^3^ on 0 and 15th days, respectively (Table 1). For NE-Cela treatment, spheroids demonstrated a diffused structure at 0.6 µM concentration on day 15, with a reduction (1.5 ± 0.1 mm^3^) in the control spheroid. In contrast, spheroids treated with plain Cela retained their structure with modest changes (2.2 ± 0.4 mm^3^). By day 12, spheroids treated with NE-Cela at 1.2 µM exhibited evidence of structural disruption, suggesting effective anti-cancer activity of NE-Cela in 3D spheroid cultures. This effect was not observed for plain Cela treatment, suggesting the superior anticancer activity of NE-Cela. A similar enhancement in anti-cancer activity was observed by Ahmad et al., where NE-mediated delivery of DHA-SBT-1214, a novel omega-3 fatty acid conjugated taxoid prodrug, demonstrated greater anticancer activity for prostate cancer than its solution counterpart [60].

##### Live–Dead Assay

To further validate the effectiveness of NE-Cela against the spheroids, we performed a live–dead fluorescent staining assay to visualize the live and dead cells in the spheroids (Appendix A). The principle of this assay is based on the staining of live cells with calcein AM, a green, fluorescent dye, while the dead cells are stained with Ethd-III, a red fluorescent dye. Corroborating data in Figure 8 show that in the multidose treatment of 1.2 µM Cela, red fluorescence was present at the core of the spheroid, suggesting dead cells and green fluorescence at the periphery correlates with living cells (Appendix A). Meanwhile, NE-Cela displayed intensified red fluorescence with little to no green fluorescence, suggesting the presence of majorly dead cells. These findings establish the improved anti-tumor effect of NE-Cela compared to its plain counterpart.

### 3.5. Stability Studies

The physical stability of NE-Cela was assessed at a storage temperature of 4 °C and accelerated conditions of 40 °C 75% relative humidity in a refrigerator and stability chamber, respectively (Appendix A). As can be seen, no significant change in percentage drug entrapment was seen in both sets at the end of 4 weeks, suggesting stability of the NE from drug leaching (Appendix A). The globule size comparison between day 0 and day 28 demonstrated a slight non-significant increase from 213.3 ± 2.8 nm to 224.6 ± 17 nm in samples kept in the refrigerator as well as in the stability chamber (Appendix A). There was no significant difference in PDI and zeta potential for both sets, suggesting stability of the NE (Appendix A). Additionally, the NE did not show phase separation or the presence of creaming. Therefore, these results demonstrate the stability of NE-Cela at the storage temperature of 4 °C and accelerated conditions of 40 °C/75% relative humidity.

## 4. Discussion

Celastrol (Cela) is a potent therapeutic agent that has shown promise in treating multiple cancers, including cervical [61], breast [62], colon [63], and various other cancers. However, its clinical translation is hindered by its low aqueous solubility of (13.3 ± 0.8 μg/mL at 37 °C) [28]. While Cela’s highly lipophilic nature enables it to penetrate cellular phospholipid membranes easily, it also presents extensive formulation and delivery challenges. Over the past decade, researchers have explored various methods to enhance Cela’s bioavailability by developing different drug delivery systems, including liposomes [64], PANAM dendrimer nanocarriers [65], and block copolymer micelles [66]. Despite these efforts, no formulation of Cela has successfully transitioned from the laboratory to clinical use.

Nanoemulsions (NE) are considered one of the most promising drug delivery systems, especially for hydrophobic drugs like Cela, due to their high miscibility in the oil phase. For instance, Lee et al. formulated an NE to deliver tanshinone, a hydrophobic herbal phytoconstituent derived from the Chinese medicinal herb *Salvia miltiorrhiza* [67]. Anticancer studies conducted using A549 cells demonstrated the effectiveness of NEs in delivering antineoplastic agents, successfully addressing the solubility challenges associated with tanshinone [67].

In the present study, the NE was composed of Capmul MCM (25%), Tween 80 (20%), Transcutol HP (5%), and water (50%) as the oil, surfactant, co-surfactant, and aqueous phase, respectively. This composition was selected as it formed a stable NE with optimum size, high encapsulation, and with the least quantity of surfactants. By employing this NE formulation, we effectively overcame the solubility issues associated with Cela, enhancing its therapeutic potential for NSCLC treatment. The choice of excipients in the formulation of NEs is crucial, as they directly impact the drug’s stability, bioavailability, and overall therapeutic efficacy. Capmul MCM comprises diglycerides of medium-chain fatty acids and has been widely reported in the literature as an oil phase in lipid-based drug delivery systems, including NEs and self-emulsifying drug delivery systems (SEDDS) [68,69]. Over the years, the role of Capmul MCM in formulations has been interchangeable as surfactants as it possesses mild surfactant properties (HLB~5–6); its primary function in our formulation is as a lipid carrier that solubilizes the drug, thereby behaving as an oil phase [70]. The biocompatibility and potential cytotoxicity of the carrier, Capmul MCM, have been evaluated in recently published studies, demonstrating its suitability for drug delivery applications [70,71]. In addition, a recent study by Elbardisy et al. also demonstrated the feasibility of Capmul MCM as an excipient for nebulized tadalafil nanoemulsion, in both in vitro and in vivo systems [72]. The surfactant is another critical component of the NE, which reduces the interfacial tension at the oil and water interface, stabilizing the oil globule. Tween 80 has a hydrophilic–lipophilic balance (HLB) value of 15 and a long aliphatic tail, rendering it lipophilic, which is ideal for O/W emulsions [73]. Moreover, Tween 80 was used to prepare nanoemulsion utilizing low-energy methods such as spontaneous emulsification [49,67]. Amani et al. prepared budesonide NE with the incorporation of Tween 80 (10% *w*/*w*), which exhibited a distinct improvement over the marketed suspension of budesonide [74]. A co-surfactant, i.e., Transcutol HP, was incorporated to provide fluidity and stability to the surfactant-coated oil globules [47,48]. Transcutol HP (HLB~4) is a water-miscible solubilizer. Although it has some lipophilic characteristics, it does not form stable emulsions independently. In the present study, Transcutol HP serves as a cosurfactant, enhancing the dispersion of Capmul MCM and reducing interfacial tension, contributing to a NE’s formation. Characterization studies of NE-Cela indicated a nanosized monodisperse system with a high encapsulation efficiency. This efficiency can be attributed to the presence of the surfactant mix, i.e., Tween 80 and Transcutol HP, which enhances the solubility of Cela for encapsulation in the oil phase.

The utility of NEs for inhaled delivery of therapeutics has been explored in recent years [75,76,77]. The inhalation route offers many advantages compared to other routes of drug administration, including allowing direct delivery to the lungs and enhancing therapeutic outcomes by maximizing local drug concentrations while minimizing systemic exposure [78]. Asmawi et al. demonstrated the effectiveness of aerosolized NEs for delivering herbal anticancer agents directly to lung tissues with deposition in the lower airways of the lungs [79]. In another study, Yang et al., developed Osimertinib-loaded NE for targeted lung cancer therapy via pulmonary delivery [75]. When coupled with ultrasound therapy, a significant decrease in tumor size was observed in H1975 orthotopic tumor-bearing mice. While promising, for inhalation therapy to be effective, the formulation must have specific respirable characteristics such as MMAD ranging from 2 to 5 µm for efficient deep lung targeting and respirable fraction (FPF) to be about 70%. To investigate this, we conducted in vitro aerosolization studies for Cela-NE using the NGI, which simulates the lung deposition of inhaled particulate matter. The NE is dispersed as small droplets, and the NGI plates, each with its cutoff diameter, simulate different parts of the respiratory system. The findings from this study, especially MMAD and FPF, confirmed the respirability and the ability of NE-Cela to deposit the dose within the deep lung tissues. In a recent study, Wang et al., achieved similar MMAD values for Cela-loaded polymeric nanoparticles, thus corroborating the findings of the current study [41]. Therefore, NE-Cela has good aerosolization characteristics and can be administered in preclinical and clinical setup via nebulization.

Once in the lung tissue, the drug must diffuse from the NE into the lung tissue. This was simulated by conducting drug release studies in a medium-mimicking lung environment. The results indicated a slow release of Cela over an extended period. Moreover, the kinetic profiling of the drug release profile revealed a zero-order drug release kinetic model. This means the drug release rate was independent of the drug’s concentration and remained constant throughout the entire release period. This sustained release characteristic of NE-Cela is beneficial in simplifying the dosage regimen of patients with NSCLC. In addition to good aerosolization characteristics, the NE must retain stability over its storage period. The nanosize of the droplets renders the NE a thermodynamically unstable system. Physical instability of NE refers to the tendency of droplets to coalesce or separate into two phases, while chemical stability refers to the degradation of the drug [80]. Therefore, over four weeks, NE-Cela was studied concurrently for its physical characteristics; i.e., globule size, surface charge, and chemical stability concerning drug content. Experimental findings revealed minimal changes in globule size, surface charge, and drug encapsulation efficiency, suggesting that NE-Cela would maintain its stability during product shelf life.

The cytotoxic effect of Cela has been extensively investigated through 2D monolayer studies, particularly in the context of NSCLC. For instance, Wang et al. reported a modest reduction in H1975 and A549 cells following 24 h treatment with Cela (0.5 μmol/L) [81]. Their study further demonstrated the in vivo efficacy of a synergistic approach targeting both cell and epidermal growth factor receptor (EGFR) pathways [81]. In another study, Jun et al. administered Cela with radio-sensitizing (RS) agents to study the combined effect of radio- and chemotherapy in a murine model of human lung carcinoma [82]. In the present study, NE-Cela alone was found to have a significant cytotoxic effect at minimal concentrations in various NSCLC cell lines as opposed to combination studies conducted in previous research. Moreover, to confirm that the cytotoxicity was due to Cela alone and not due to components of the NE, representative non-malignant cells (HEK-293, AEC) were treated with blank NE at the same concentration as that of NE-Cela. The results from this study demonstrated no significant cytotoxic effect, thus confirming the biocompatibility of the blank NE. Although the excipients used in the preparation of the NE—Capmul MCM, Tween 80, and Transcutol HP—are not FDA-approved for inhalation, the biocompatibility results support the safety profile of these excipients. Capmul MCM could solubilize 10 mg Cela (4.8 mg/mL), a sufficient dose for inhalation. It is also classified as generally recognized as safe (GRAS) and has been used in numerous intranasal formulations [83]. Tween 80 has also been used as a surfactant in NE formulations for intranasal administration (WO2020129085A1). However, Lindenberg et al. observed some lung toxicity of Tween 80 at high concentrations (10 *v*/*v* %) using an air–liquid interface (ALI) and liquid/liquid (L/L) exposure models [84]. In the current formulation, NE-Cela was diluted 50-fold for nebulization, reducing the Tween 80 to 0.4% concentration, which remains within acceptable limits. Preclinical inhalation studies on Transcutol HP have shown mild local irritation in the rat larynx. However, no adverse effects were observed in mice, guinea pigs, rabbits, or cats following daily exposure to a Transcutol-saturated atmosphere for 12 days [85]. Elbardisy et al. demonstrated the safety of Capmul-MCM-EP containing nanoemulsions in both in vitro and oro-tracheally administered in vivo in rat studies [72]. This literature evidence supports the use of these excipients; however, further preclinical inhalation studies are needed to confirm their inhalation safety.

Cellular proliferation and migration play crucial roles in the beginning and progression of different types of cancers. Several studies have highlighted the inhibitory potential of Cela in various cancer cell lines. For instance, Lee et al. reported Cela-induced inhibition of cell migration and increased G1 arrest in gastric cancer cell-cycle populations [86]. Similarly, Guo et al. reported the suppression of colorectal cancer cells via nitric oxide activity and the inhibition of angiogenesis [87]. Lin et al. explored the apoptotic activity of Cela on mitochondrial and Fas-mediated pathways in head and neck cancer cells [88]. Another study identified histone acetylation associated with the NuA4 histone acetyltransferase complex as being involved in the anti-proliferation actions of Cela in treating lung cancer [89]. As the mechanism of action of Cela is well-established in cancer cells, and the current formulation (NE-Cela) is not expected to enable a new mechanism, further mechanistic studies were not conducted. Additionally, previous studies from our group have evaluated the effects of nanoencapsulation on Cela’s cellular activity [41]. These studies demonstrated that encapsulating Cela in PLGA microparticles did not significantly alter the antioxidant and apoptotic activity of Cela in mesothelioma cancer cells. These findings confirm that while nanoencapsulation enhances Cela’s stability and release profile, it does not modify its underlying mechanisms of action in cancer cells. In the present study, we investigated the effect of NE-Cela on NSCLC. Scratch and clonogenic assays were utilized to assess the ability of NE-Cela to hinder cell–cell interaction and discourage the development of tumors from individual NSCLC cells. These studies also revealed the potential of NE-Cela in preventing tumor recurrence through colony formation over an extended period.

Three-dimensional cell-based assays closely resemble in vivo tissue architecture, providing more physiologically relevant models of diseases [90]. Moreover, spheroids allow for complex cell–cell interaction that mimics the microenvironment of tumors. This interaction is essential for understanding cellular behavior and the impact of cell interactions on disease progression. In a recent study, Parvathaneni et al. employed a 3D cell assay to investigate the effect of amodiaquine-loaded, folic acid-conjugated polymeric nanoparticles on breast cancer cells [91]. Their findings revealed a reduction in tumor size in response to the drug-loaded nanoparticles. Similarly, Wang et al. performed a 3D spheroid assay to assess the treatment of Cela-loaded polymeric particles for treating malignant pleural mesothelioma [41]. Their findings demonstrated superior anti-tumor activity when compared to Cela only. A recent study demonstrated the enhanced anticancer activity of resveratrol-cyclodextrin-loaded nanoparticles in 3D NSCLC tumors compared to plain resveratrol [43]. In the present paper, a similar 3D spheroid study was performed with two dosage regimes, i.e., single and multiple doses. Findings demonstrated a significant reduction in tumor volume in both treatment groups, with the spheroid mass disintegrating on day 12 in the multidose treatment group.

## 5. Conclusions

This study demonstrates that a stable Cela-loaded NE can be prepared using a low-energy emulsification water titration technique. Effective Cela-loaded lipid-based NE pulmonary delivery via inhalation may be possible by reducing Cela dosage, dosing frequency, and associated toxicities. The optimized formulation has favorable physiochemical characteristics and can be administered via the pulmonary route to tackle non-small cell lung cancer. Moreover, the drug is released in a sustained manner over a long period. In vitro studies indicated the efficacy of Cela-loaded NE in enhancing anticancer activity in both monolayers and spheroid cancer models. Stability studies showed the stability of the formulation in storage and at accelerated conditions. These promising results can be used as a framework for designing preclinical and potential clinical studies to treat non-small cell lung cancer.

## Figures and Tables

**Figure 1 pharmaceutics-17-00540-f001:**
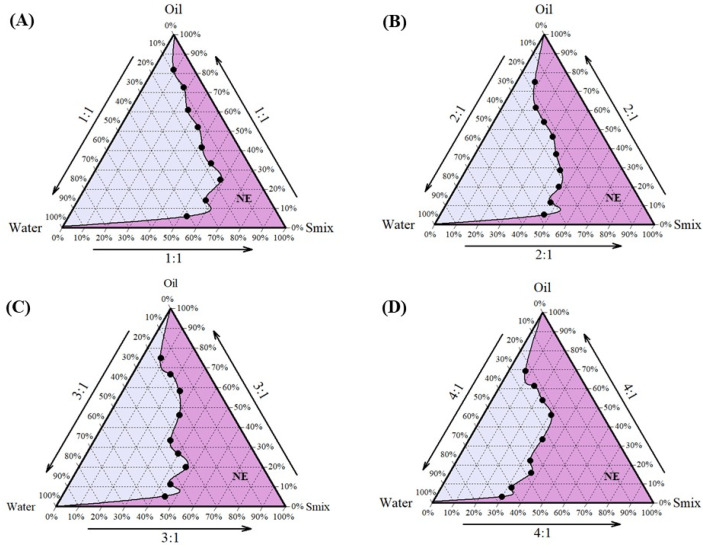
Preformulation studies for development of NE-Cela: pseudo-ternary phase diagrams of S_mix_ ratios (**A**) 1:1, (**B**) 2:1, (**C**) 3:1, and (**D**) 4:1. The pseudo-ternary phase diagram was prepared by titrating water with different proportions of oil and S_mix_. S_mix_ comprised different ratios of surfactant and co-surfactant i.e., Tween 80 and Transcutol HP. The dark pink region represents the composition forming clear NE (*n* = 3).

**Figure 2 pharmaceutics-17-00540-f002:**
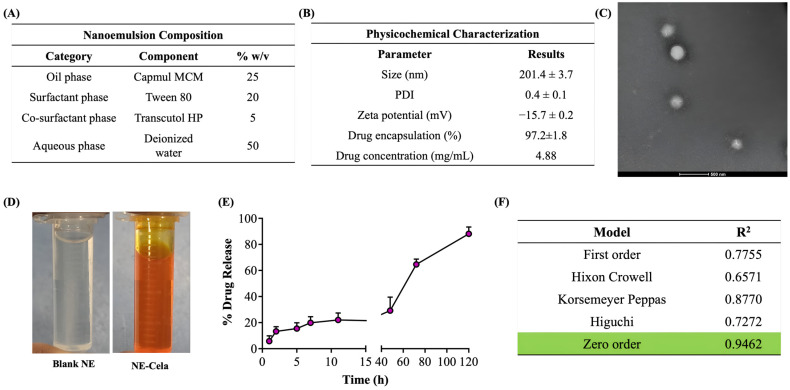
(**A**) Composition of NE-Cela. (**B**) Characterization studies of NE-Cela, (**C**) TEM analysis at 50× dilution (the scale bar for images represents 500 nm), and (**D**) pictorial representation of blank NE and NE-Cela. (**E**) In vitro drug release profile for NE-Cela in simulated lung fluid using the tube method. (**F**) Drug release data fitted to various kinetic models with the corresponding R^2^ value. Data represent mean ± SD (*n =* 3).

**Figure 3 pharmaceutics-17-00540-f003:**
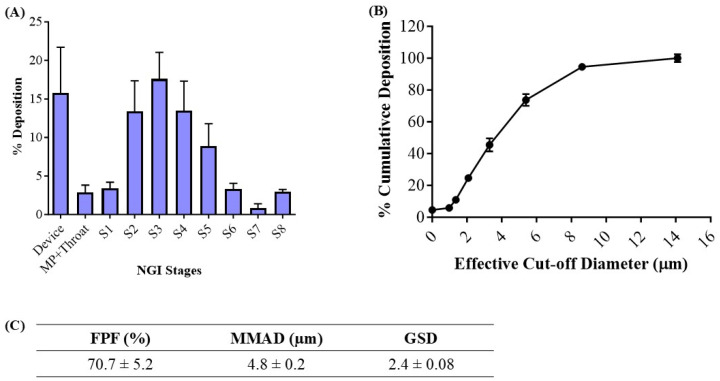
In vitro aerosolization results for NE-Cela. (**A**) Deposition pattern in the NGI stages 1–8, (**B**) cumulative mass deposition as a function of the NGI effective cut-off diameters and (**C**) in vitro aerosolization parameters. Data represent mean ± SD (*n =* 3).

**Figure 4 pharmaceutics-17-00540-f004:**
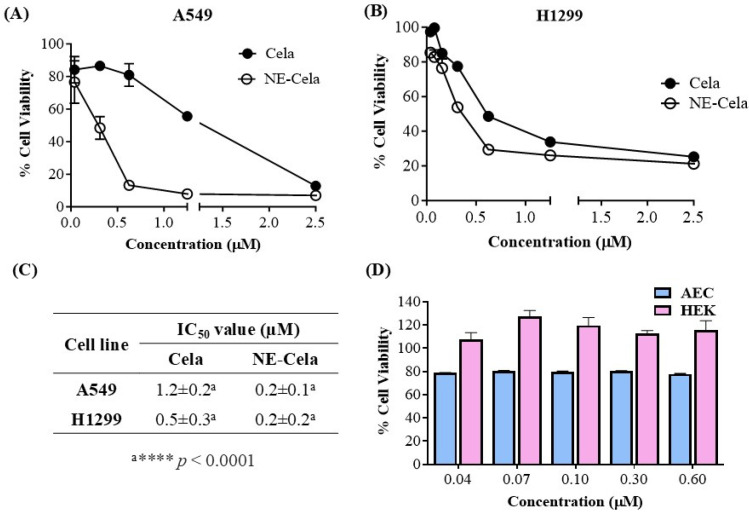
Inhibitory effects on different cell lines (**A**) A549 and (**B**) H1299, after treatments with Cela and NE-Cela. Cell viability in (**C**) IC_50_ of NE-Cela and Cela in A549 and H1299 and (**D**) HEK-293 and AEC normal cell lines after treatment with blank NE. Data represent mean ± SD (*n =* 6) of at least 3 independent trials.

**Figure 5 pharmaceutics-17-00540-f005:**
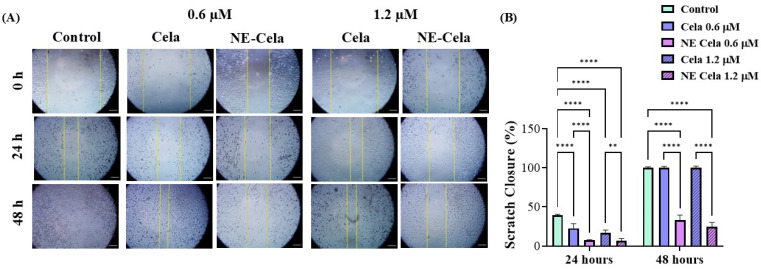
In vitro scratch wound healing assay with A549 cells treated with Cela and NE-Cela with no treatment as a control. (**A**) Representative images for indicated treatments taken at 10× magnification using LAXCO microscope. (**B**) Quantitative estimation of % scratch closure over 24 and 48 h. A monolayer of cancer cells was grown followed by the application of scratch in each well. The scratched cells were treated with NE-Cela and Cela and imaged at 0 and 48 h to observe the closure of the scratch. The uncovered area was quantified by the ImageJ Software (v1.54) at each time point represented in the graphs as mean ± SEM of three different experiments. Ne-Cela significantly inhibited the closure of scratches at both 24 and 58. **** *p* < 0.0001, ** *p* < 0.01.

**Figure 6 pharmaceutics-17-00540-f006:**
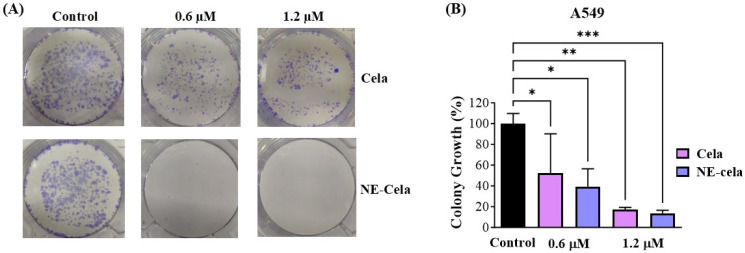
Colony forming assay. (**A**) Representative images showing distinct colonies of A549 after staining with crystal violet. (**B**) Qualitative estimation of % colony growth. A549 cells were seeded at low density in 6 well plates and were treated with Cela and NE-Cela at two concentrations. Treatments were replaced with fresh media for another 7 days with media replacement on alternative days. Colonies were then washed with PBS, fixed with 4% paraformaldehyde, and then stained with crystal violet, and photographed. The images were then quantified using colony counting software (OpenCFU 3.8-BETA). Significance was established by one-way ANOVA and Dunnet’s multiple comparison test, *** *p* < 0.005, ** *p* < 0.005, * *p* < 0.05. Data represent mean ± SD (*n =* 3).

**Figure 7 pharmaceutics-17-00540-f007:**
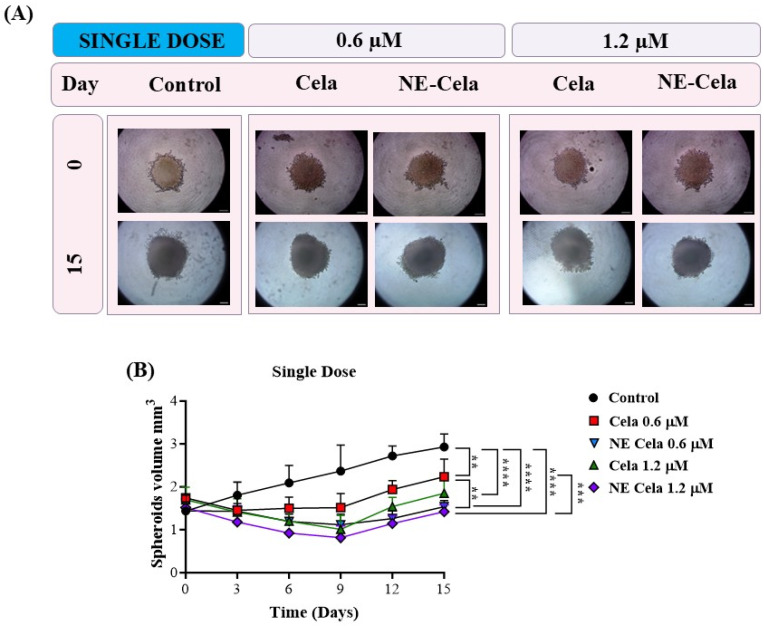
A 3D cell culture study was employed to investigate the effect of NE-Cela in a physiologically relevant culture model. A549 cells were seeded in the U bottom plate to initiate the formation of the spheroid. The spheroids were treated with NE-Cela and its plain counterpart at two dosage regimens; i.e., single, and multiple doses. Microscopic images of spheroid study. (**A**) Single dose show a significant decrease in tumor volume. (**B**) The scale bar for the images represents 200 μm. Data represent mean ± SD (*n =* 6) and significance between the groups was analyzed by one-way ANOVA and Tukey’s multiple comparison test. **** *p* < 0.0001, *** *p* < 0.0005, ** *p* < 0.005.

**Figure 8 pharmaceutics-17-00540-f008:**
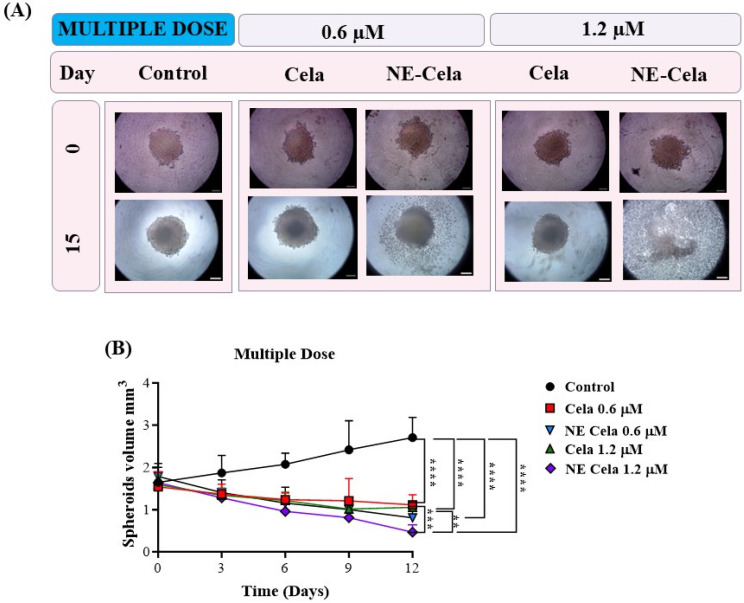
The effects of multidose treatment on spheroids. (**A**) Microscopic pictures and (**B**) spheroid volume analysis on different days results suggest a significant decrease in tumor volume in a multidose regimen. The spheroid treated with NE-Cela completely disintegrated between day 12 and day 15. The scale bar for the images represents 200 μm. Data represent mean ± SD (*n =* 6) and significance between the groups was analyzed by one-way ANOVA and Tukey’s multiple comparison test. **** *p* < 0.0001, *** *p* < 0.0005, ** *p* < 0.005.

**Table 1 pharmaceutics-17-00540-t001:** Percentage change in spheroid volume after single and multiple dose treatments with Cela and NE-Cela at two concentrations. Data represent mean ± SD (*n =* 6).

**Spheroid Volume (mm^3^)**		**0.6 μM**		**1.2 μM**	
**Single Dose**	**Control**	**Cela**	**NE-Cela**	**Cela**	**NE-Cela**
Day 0	1.4 ± 0.3	1.7 ± 0.1	1.4 ± 0.3	1.7 ± 0.1	1.5 ± 0.2
Day 15	2.9 ± 0.9	2.2 ± 0.4	1.5 ± 0.1	1.8 ± 0.3	1.4 ± 0.1
% Volume change	107.1 ± 40.4	29.4 ± 5.6	7.1 ± 1.6	5.9 ±1.0	−6.7 ± 1.01
**Multiple Dose**	**Control**	**Cela**	**NE-Cela**	**Cela**	**NE-Cela**
Day 0	1.6 ± 0.3	1.5 ± 0.3	1.7 ± 0.3	1.6 ± 0.2	1.6 ± 0.2
Day 15	2.7 ± 0.4	1.1 ± 0.2	0.8 ± 0.1	1.0 ± 0.1	-
% Volume change	68.7 ± 16.4	−26.7 ± 7.2	−52.9 ± 11.5	−37.5 ± 6.0	-

## Data Availability

The data will be made available on request.

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
