# Peer review of "An Inhaled Nanoemulsion Encapsulating a Herbal Drug for Non-Small Cell Lung Cancer (NSCLC) Treatment"

_pharmaceutics, 2025, doi:10.3390/pharmaceutics17050540_

Round 1
Reviewer 1 Report
Comments and Suggestions for Authors
- Regarding the UHPLC data in Fig. S1, Fig. S1A only qualitatively presents the absorption peak of a single sample, while Fig. S1B provides a complete standard curve containing information on six samples. It is recommended to include the UHPLC absorption peak graphs of these six samples in Fig. S1A.
- It is suggested to include the fitted equation in Fig. S1B.
- The biocompatibility and potential cytotoxicity of the carrier (Capmul MCM) should be evaluated, or relevant literature should be cited. Moreover, some smart nanocarriers for herbal drug delivery should be cited, such as Ind. Crops Prod., 2025, 228, 120902.
- The inhaled NE-Cela formulation must pass through the respiratory tract to reach the lungs. However, the manuscript does not include a safety assessment of drug administration via the respiratory route. It is recommended to provide such an evaluation or cite relevant literature.
- What are the advantages of NE-Cela administered via inhalation compared to direct intravenous injection?
- In the cytotoxicity evaluation, a significant difference in toxicity between NE-Cela on A549 cells and H1299 cells was observed. Please explain the reason—does this result from the selectivity of the carrier or the drug?
None
Author Response
REVIEWER #1
1. Regarding the UHPLC data in Fig. S1, Fig. S1A only qualitatively presents the absorption peak of a single sample, while Fig. S1B provides a complete standard curve containing information on six samples. It is recommended to include the UHPLC absorption peak graphs of these six samples in Fig. S1A:
Reply: We appreciate the reviewer’s suggestion. In response, we have overlaid the UHPLC chromatograms of the linearity samples in Fig. S1A to provide a more comprehensive representation of the data.
2. It is suggested that the fitted equation be included in Fig. S1B.
Reply: We appreciate the reviewer’s suggestion. In response, we have included the fitted equation in Fig. S1B to provide a clearer representation of the data and enhance the understanding of the standard curve.
3. The biocompatibility and potential cytotoxicity of the carrier (Capmul MCM) should be evaluated, or relevant literature should be cited. Moreover, some smart nanocarriers for herbal drug delivery should be cited, such as Ind. Crops Prod., 2025, 228, 120902.
Reply: We appreciate the reviewer’s suggestion. The biocompatibility and potential cytotoxicity of Capmul MCM, the carrier used in this study, have been evaluated in prior research, demonstrating its suitability for drug delivery applications. The manuscript references these studies (lines 536-540, Ref 70-72). While biocompatibility studies of Capmul MCM as an individual component were specifically not conducted in this study, we have included biocompatibility of the blank nanoemulsion in AEC and HEK cells. Additionally, recent studies on smart nanocarriers for herbal drug delivery, such as those discussed in Ind. Crops Prod., 2025, 228, 120902, have been cited to further support the potential of nanocarriers in enhancing the delivery and efficacy of herbal drugs (lines 91-94, Ref 32).
4. The inhaled NE-Cela formulation must pass through the respiratory tract to reach the lungs. However, the manuscript does not include a safety assessment of drug administration via the respiratory route. It is recommended to provide such an evaluation or cite relevant literature.
Reply: We thank the reviewer for this important suggestion. We acknowledge the necessity of assessing the safety of drug administration via the respiratory route. Although the manuscript does not present specific safety data for NE-Cela via inhalation, we have conducted preliminary safety evaluations using primary human AEC (alveolar epithelial cells). These studies demonstrated that NE-Cela did not exhibit significant cytotoxicity at therapeutic concentrations (lines 364-372). In addition, Elbardisy et al. demonstrated both in-vitro and in-vivo safety of tadalafil nanoemulsions with similar components with Capmul-MCM-EP as oil phase, following oro-tracheal administration (line 538-540, Ref 72).
5. What are the advantages of NE-Cela administered via inhalation compared to direct intravenous injection?".
Reply: We thank the reviewer for this important question. Inhalation delivery of NE-Cela offers several advantages over intravenous (IV) administration. These include direct drug deposition at the disease site, reduced systemic exposure, and minimized off-target toxicity. Additionally, pulmonary delivery can bypass first-pass metabolism and potentially allow for lower doses to achieve therapeutic efficacy. (discussed in the manuscript, line109-114).
6. In the cytotoxicity evaluation, a significant difference in toxicity between NE-Cela on A549 cells and H1299 cells was observed. Please explain the reason—does this result from the selectivity of the carrier or the drug?
Reply: We thank the reviewer for this insightful comment. The observed difference in cytotoxicity between NE-Cela on A549 and H1299 cells is likely attributed to the intrinsic differences between the two cell lines rather than the selective activity of the nanoemulsion (NE) carrier. H1299 and A549 are both non-small cell lung cancer (NSCLC) cell lines; however, H1299 cells are more aggressive and generally more sensitive to treatment compared to A549 cells. Although both cell lines harbor similar mutations, such as KRAS, a key distinction is that A549 cells express functional p53, while H1299 cells lack this tumor suppressor. These biological differences may contribute to the variation in response to Cela. Moreover, the blank NE did not show significant cytotoxicity in either cell line, suggesting that the observed activity is primarily drug-driven. These points have been addressed in the revised manuscript (lines 359-362, Ref 53).

Reviewer 2 Report
Comments and Suggestions for Authors
In this manuscript, a Celastrol-loaded nanoemulsion (Cela-NE) was designed for inhalational delivery in the treatment of non-small cell lung cancer (NSCLC). This manuscript is well organized and presents promising in vitro results, indicating the use of 3D tumor spheroids models. However, there are several aspects require clarification or further experimental support.
- In the abstract, authors mention the “penetrability” of NE-Cela. However, this is not sufficiently demonstrated in the results. Authors need to include the penetration assay (e.g., cellular uptake in the 3D spheroids, confocal microscopy, or flow cytometry) to confirm this claim.
- The formula labeled as Eq.1 is incorrect for calculating of EE. %EE should be (the amount of drug entrapped)/(total drug added)X100. Please revise this accordingly. Please also update the numbers.
- The manuscript should clearly state the formula used to calculate the cumulative drug release in the in vitro release studies. This is essential for reproducibility and transparency.
- While the cytotoxicity results are promising, the authors should consider including flow cytometry-based apoptosis assays (e.g., Annexin V/PI staining) to quantify the percentage of apoptotic vs. viable cells. This would complement the live-dead assay and strengthen the biological evaluation of NE-Cela.
- In figure 4D, the AEC cell group does not display error bars. Please clarify whether this was due to a single replicate or a plotting error. If replicates were performed, error bars should be included.
- In figure 5A, the scratch wound images at 0 h are inconsistent across treatment groups. Please explain why the initial scratch areas are not visually similar, and confirm whether normalization or quantitative correction was applied during analysis.
- In figure 7, there appears to be little to no difference in tumor volumes across all treated groups on Day 15. This seems inconsistent with the earlier claims of efficacy. Please clarify or provide additional statistical analysis to support the conclusions.
- The authors acknowledge that the excipients are not FDA-approved for inhalation. It would be better to include some supporting inhalation toxicity data.
- The authors mention that Celastrol’s mechanism is well-established and therefore mechanistic studies were not performed. However, given the nanoemulsion formulation, it would be valuable to at least discuss whether nanoencapsulation could alter cellular uptake or pathway activation compared to free Celastrol.
Author Response
REVIEWER #2
In this manuscript, a Celastrol-loaded nanoemulsion (Cela-NE) was designed for inhalational delivery in the treatment of non-small cell lung cancer (NSCLC). This manuscript is well organized and presents promising in vitro results, indicating the use of 3D tumor spheroids models. However, there are several aspects require clarification or further experimental support.
- In the abstract, authors mention the “penetrability” of NE-Cela. However, this is not sufficiently demonstrated in the results. Authors need to include the penetration assay (e.g., cellular uptake in the 3D spheroids, confocal microscopy, or flow cytometry) to confirm this claim.
Reply: We thank the reviewers for this valuable suggestion. We hypothesize that the enhanced cytotoxic effect of NE-Cela may be attributed to its improved penetrability. However, as the drug lacks inherent fluorescence and we currently do not have access to confocal microscopy, a penetration assay could not be performed. Accordingly, the term “enhanced penetrability” has been revised to “enhanced activity” in the abstract (line 30) and results section (line 475-481). Additionally, appropriate references have been included to support the hypothesis of improved permeation (line 481, Ref 60).
The formula labeled as Eq.1 is incorrect for calculating of EE. %EE should be (the amount of drug entrapped)/(total drug added)X100. Please revise this accordingly. Please also update the numbers.
Reply: We thank the reviewer for bringing this to our attention. The calculation of %EE was done according to the reviewer's suggestion, i.e., amt of drug quantified/ amt of drug added x 100. However, the formula mentioned in the manuscript was erroneous and has been rectified (line 186).
The manuscript should clearly state the formula used to calculate the cumulative drug release in the in vitro release studies. This is essential for reproducibility and transparency.
Reply: We appreciate the reviewer’s suggestion. In this study, we utilized a multiple-tube method for evaluating in vitro drug release, wherein separate tubes containing the formulation were incubated under identical conditions. At each predetermined time point, one tube was withdrawn and analyzed for drug content without replacement or pooling. This approach allowed for the quantification of the drug released at each time point independently. As such, the release data presented represent non-cumulative, time-point-specific values, and no cumulative release calculation or formula was applied. This methodology has now been clarified in the Materials and Methods section (section 2.5.4, line 190-195) to improve transparency and reproducibility.
While the cytotoxicity results are promising, the authors should consider including flow cytometry-based apoptosis assays (e.g., Annexin V/PI staining) to quantify the percentage of apoptotic vs. viable cells. This would complement the live-dead assay and strengthen the biological evaluation of NE-Cela.
Reply: We thank the reviewer for this insightful suggestion. We agree that flow cytometry-based apoptosis assays such as Annexin V/PI staining provide a robust evaluation of apoptotic profiles. However, as noted in our previous work, we have already investigated the apoptotic effects of nano-encapsulated Celastrol (PLGA-Cela) in mesothelioma cells using mechanistic assays including caspase-3 activation, which confirmed that nanoencapsulation preserves Celastrol’s apoptotic activity (Ref 90). These studies showed no significant alteration in apoptosis pathway activation when compared to free Celastrol. Based on these prior findings, the current study focused on evaluating the therapeutic efficacy of the inhalable NE-Cela formulation. While we acknowledge the value of additional apoptosis quantification via flow cytometry, we did not repeat these assays in the present work to avoid redundancy. This rationale has now been incorporated into the revised manuscript (lines 634-639) for clarity.
- In figure 4D, the AEC cell group does not display error bars. Please clarify whether this was due to a single replicate or a plotting error. If replicates were performed, error bars should be included.
Reply: We thank the reviewer for pointing this out. The absence of error bars in the AEC cell group in Fig. 4D was due to a plotting error. This has been rectified, and the error bars for the AEC cell group are now included in the revised figure (Fig. 4D).
- In Figure 5A, the scratch wound images at 0 h are inconsistent across treatment groups. Please explain why the initial scratch areas are not visually similar, and confirm whether normalization or quantitative correction was applied during analysis.
Reply: We thank the reviewer for this valuable observation. The slight variation in the initial scratch width across treatment groups in Figure 5A is attributed to manual pipette tip-based scratching, which can result in minor inconsistencies in initial wound areas. However, to account for these differences and ensure accurate comparisons, the percent scratch closure was calculated for each group relative to its own 0 h baseline. This approach allows the analysis to reflect the relative migration capacity of cells over time, irrespective of initial wound size, ensuring data normalization across groups. These methodological details have been clarified in the manuscript (lines 386-387, Section 3.4.2)
In figure 7, there appears to be little to no difference in tumor volumes across all treated groups on Day 15. This seems inconsistent with the earlier claims of efficacy. Please clarify or provide additional statistical analysis to support the conclusions.
Reply: We thank the reviewer for this important observation. While the qualitative differences in spheroid size between treatment groups in Figure 7A may appear to be modest, quantitative image analysis revealed statistically significant variations in tumor volume, particularly in the NE-Cela-treated groups. Specifically, the longest spheroid diameter was measured using ImageJ, and tumor volume was subsequently estimated using the formula for a sphere: Volume=3/4πr3, where r is half the measured diameter. This method enabled precise volumetric assessment beyond visual inspection. The results demonstrated a reduction in tumor volume in the NE-Cela (1.2 µM) group relative to the control and Cela-treated groups, supporting the enhanced efficacy of the nanoemulsion formulation. Statistical analysis is provided in the legends for Figs. 7 & 8.
The authors acknowledge that the excipients are not FDA-approved for inhalation. It would be better to include some supporting inhalation toxicity data.
Reply: We appreciate the reviewer’s insightful comment. Supporting information regarding the inhalation toxicity of polysorbate 80 has been added to the manuscript (line 612-614, Ref 85). Specifically, we have referenced findings by Lindenberg et al., who reported lung toxicity at high concentrations (10% v/v) using ALI and L/L models. Multiple other safety studies cited demonstrate the safety of other excipients in various in-vivo models of inhalation (lines 612-621, Ref 85, 86, 72). However, in our study, the NE-Cela formulation was diluted 50-fold for nebulization, reducing the polysorbate 80 concentration to 0.4%, which is within acceptable safety limits. This provides context and justification for the safe use of the excipient at the concentrations employed in our formulation. These points have been added to the revised manuscript.
- The authors mention that Celastrol’s mechanism is well-established and therefore mechanistic studies were not performed. However, given the nanoemulsion formulation, it would be valuable to at least discuss whether nanoencapsulation could alter cellular uptake or pathway activation compared to free Celastrol.
Reply: We thank the reviewer for highlighting the importance of discussing whether nanoencapsulation could alter cellular uptake or pathway activation compared to free Celastrol. Previous studies from our laboratory have addressed these concerns. In these studies, the cellular uptake and mechanistic pathway activation of nano-encapsulated Celastrol (PLGA-Cela) was compared to free Celastrol in mesothelioma cancer cells. The results demonstrated that PLGA-Cela did not alter the induction of key apoptotic markers, such as e.g. caspase-3 activation. These findings support the notion that nanoencapsulation, while improving Celastrol’s stability and sustained release, does not modify its established mechanisms of action. Therefore, in the current study, we focused on the formulation’s efficacy and feasibility for inhalation, and did not repeat these mechanistic evaluations. We have modified the manuscript to include this information (lines 634-639).

Round 2
Reviewer 2 Report
Comments and Suggestions for Authors
- The authors should perform additional experiments to substantiate the claim that NE-Cela has enhanced penetrability. Currently, this assertion is made without direct supporting data. A suggested approach would be to evaluate the uptake of Cela in 3D tumor spheroids. This can be achieved by growing uniform spheroids, treating them with the same concentration of either naked Cela or NE-Cela for 2 h or 4 h, dissociating the spheroids into single cells (e.g., using trypsin), counting the viable cells, and quantifying intracellular drug content via UPLC or another suitable method. The uptake should then be normalized to cell number to determine whether NE-Cela significantly improves cellular uptake compared to naked Cela.
- In the text accompanying Figure 7, the authors state: “While the qualitative differences in spheroid size between treatment groups in Figure 7A may appear to be modest, quantitative image analysis revealed statistically significant variations in tumor volume, particularly in the NE-Cela-treated groups.” Unsure is correct or not. In addition, the formula used to calculate spheroid volume should be stated and should include both the longest diameter and the vertical diameter (e.g., Volume = [Length × Width²]/2), which provides a more accurate estimate of 3D spheroid size.
Author Response
REVIEWER #2
The authors should perform additional experiments to substantiate the claim that NE-Cela has enhanced penetrability. Currently, this assertion is made without direct supporting data. A suggested approach would be to evaluate the uptake of Cela in 3D tumor spheroids. This can be achieved by growing uniform spheroids, treating them with the same concentration of either naked Cela or NE-Cela for 2 h or 4 h, dissociating the spheroids into single cells (e.g., using trypsin), counting the viable cells, and quantifying intracellular drug content via UPLC or another suitable method. The uptake should then be normalized to cell number to determine whether NE-Cela significantly improves cellular uptake compared to naked Cela.
Reply: We appreciate the reviewer’s suggestion. We agree that using 3D tumor spheroids to study the uptake of NE-Cela would provide more substantial evidence for its penetrability. However, we kindly mention that adding these experiments is outside the current scope of our work. In this study, our primary goal was to develop an inhaled NE-Cela formulation and show its initial effectiveness in-vitro. To support this, we have included references to previous studies where nanoemulsion-based formulations have demonstrated improved outcomes compared to free drugs, such as the work by Ahmad et al. on DHA-SBT-1214 (Ref 60). We believe this sets a foundation for more detailed studies in the future, including experiments like the one suggested. To address the reviewer’s point, we have removed the specific claim about enhanced penetrability from the manuscript. Instead, we now refer to the observed effect as improved efficacy. We fully agree that spheroid-based uptake study is an important aspect worth exploring, and we plan to investigate it further in follow-up studies.
In the text accompanying Figure 7, the authors state: “While the qualitative differences in spheroid size between treatment groups in Figure 7A may appear to be modest, quantitative image analysis revealed statistically significant variations in tumor volume, particularly in the NE-Cela-treated groups.” Unsure is correct or not. In addition, the formula used to calculate spheroid volume should be stated and should include both the longest diameter and the vertical diameter (e.g., Volume = [Length × Width²]/2), which provides a more accurate estimate of 3D spheroid size."
Reply: Thank you for your helpful comment. We agree that including the specific formula for calculating spheroid volume improves transparency and clarity. In response, we have now added the formula to the supplementary section along with a table summarizing the measured dimensions and calculated volumes for each group (Table S1).
